# In Situ Analyses of Placental Inflammatory Response to SARS-CoV-2 Infection in Cases of Mother–Fetus Vertical Transmission

**DOI:** 10.3390/ijms25168825

**Published:** 2024-08-13

**Authors:** Denise Morotti, Silvia Tabano, Gabriella Gaudioso, Tatjana Radaelli, Giorgio Alberto Croci, Nicola Bianchi, Giulia Ghirardi, Andrea Gianatti, Luisa Patanè, Valeria Poletti de Chaurand, David A. Schwartz, Mohamed A. A. A. Hagazi, Fabio Grizzi

**Affiliations:** 1Pathology Department, ASST Papa Giovanni XXIII, 24127 Bergamo, Italy; 2Foundation IRCCS Ca’ Granda Ospedale Maggiore Policlinico, 20122 Milano, Italygiorgio.croci@policlinico.mi.it (G.A.C.); 3Department of Pathophysiology and Transplantation, University of Milan, 20122 Milano, Italy; 4Department of Obstetrics and Gynecology, ASST Papa Giovanni XXIII, 24127 Bergamo, Italy; 5Perinatal Pathology Consulting, Atlanta, GA 30329, USA; davidalanschwartz@gmail.com; 6Department of Immunology and Inflammation, IRCCS Humanitas Research Hospital, 20089 Rozzano, Italy; mohamed.ahmedahmedabdelazizhegazi@humanitasresearch.it (M.A.A.A.H.); fabio.grizzi@humanitasresearch.it (F.G.); 7Department of Biomedical Sciences, Humanitas University, 20072 Pieve Emanuele, Italy

**Keywords:** COVID-19, SARS-CoV-2 vertical transmission, placenta, pregnancy, twins

## Abstract

It has been shown that vertical transmission of the SARS-CoV-2 strain is relatively rare, and there is still limited information on the specific impact of maternal SARS-CoV-2 infection on vertical transmission. The current study focuses on a transcriptomics analysis aimed at examining differences in gene expression between placentas from mother–newborn pairs affected by COVID-19 and those from unaffected controls. Additionally, it investigates the in situ expression of molecules involved in placental inflammation. The Papa Giovanni XXIII Hospital in Bergamo, Italy, has recorded three instances of intrauterine transmission of SARS-CoV-2. The first two cases occurred early in the pandemic and involved pregnant women in their third trimester who were diagnosed with SARS-CoV-2. The third case involved an asymptomatic woman in her second trimester with a twin pregnancy, who unfortunately delivered two stillborn fetuses due to the premature rupture of membranes. Transcriptomic analysis revealed significant differences in gene expression between the placentae of COVID-19-affected mother/newborn pairs and two matched controls. The infected and control placentae were matched for gestational age. According to the Benjamani–Hochberg method, 305 genes met the criterion of an adjusted *p*-value of less than 0.05, and 219 genes met the criterion of less than 0.01. Up-regulated genes involved in cell signaling (e.g., CCL20, C3, MARCO) and immune response (e.g., LILRA3, CXCL10, CD48, CD86, IL1RN, IL-18R1) suggest their potential role in the inflammatory response to SARS-CoV-2. RNAscope^®^ technology, coupled with image analysis, was utilized to quantify the surface area covered by SARS-CoV-2, ACE2, IL-1β, IL-6, IL-8, IL-10, and TNF-α on both the maternal and fetal sides of the placenta. A non-statistically significant gradient for SARS-CoV-2 was observed, with a higher surface coverage on the fetal side (2.42 ± 3.71%) compared to the maternal side (0.74 ± 1.19%) of the placenta. Although not statistically significant, the surface area covered by ACE2 mRNA was higher on the maternal side (0.02 ± 0.04%) compared to the fetal side (0.01 ± 0.01%) of the placenta. IL-6 and IL-8 were more prevalent on the fetal side (0.03 ± 0.04% and 0.06 ± 0.08%, respectively) compared to the maternal side (0.02 ± 0.01% and 0.02 ± 0.02%, respectively). The mean surface areas of IL-1β and IL-10 were found to be equal on both the fetal (0.04 ± 0.04% and 0.01 ± 0.01%, respectively) and maternal sides of the placenta (0.04 ± 0.05% and 0.01 ± 0.01%, respectively). The mean surface area of TNF-α was found to be equal on both the fetal and maternal sides of the placenta (0.02 ± 0.02% and 0.02 ± 0.02%, respectively). On the maternal side, ACE-2 and all examined interleukins, but not TNF-α, exhibited an inverse mRNA amount compared to SARS-CoV-2. On the fetal side, ACE-2, IL-6 and IL-8 were inversely correlated with SARS-CoV-2 (r = −0.3, r = −0.1 and r = −0.4, respectively), while IL-1β and IL-10 showed positive correlations (r = 0.9, *p* = 0.005 and r = 0.5, respectively). TNF-α exhibited a positive correlation with SARS-CoV-2 on both maternal (r = 0.4) and fetal sides (r = 0.9) of the placenta. Further research is needed to evaluate the correlation between cell signaling and immune response genes in the placenta and the vertical transmission of SARS-CoV-2. Nonetheless, the current study extends our comprehension of the molecular and immunological factors involved in SARS-CoV-2 placental infection underlying maternal–fetal transmission.

## 1. Introduction

The COVID-19 pandemic, recognized as a significant public health emergency since March 2020 [1,2,3], remains impactful. Extensive endeavors in both clinical and research domains have been devoted to comprehending the dynamics of SARS-CoV-2 transmission and its repercussions on human health [4]. Yet there remain unresolved facets necessitating further exploration. Our engagement in maternal–fetal medicine commenced early due to our hospital’s proximity to the COVID-19 epicenter in Northern Italy. Swiftly, we documented two instances of SARS-CoV-2-infected pregnant women in their third trimester, both exhibiting infected placentas [5]. Using RNAscope^®^ technology, we identified SARS-CoV-2 RNA within the syncytiotrophoblast [5]. Over the past two years, original research and meta-analytical data have emerged that have provided additional insights into risk factors and mechanisms for the intrauterine transmission of SARS-CoV-2. Pregnant women encounter an elevated risk for developing severe COVID-19 compared to their infected non-pregnant counterparts [6,7]. Moreover, heightened rates of viral infection have been noted among pregnant individuals of Black, Asian, and Hispanic backgrounds, with approximately 15% experiencing preterm birth. Fortunately, occurrences of neonatal SARS-CoV-2 infections seem restricted, presenting few worrisome clinical manifestations, although more severe phenotypes and perinatal deaths have been reported [8]. In 2022, we provided evidence of the vertical transmission of SARS-CoV-2 and interstitial pneumonia in a second-trimester twin stillbirth from an infected but asymptomatic woman [9]. This underscores the risk of adverse fetal outcomes linked to SARS-CoV-2 infection during the second trimester, regardless of symptom severity. The immunohistological and molecular analyses of these four infected placentas have led us to conclude that transplacental transmission is associated with chronic histiocytic intervillositis and syncytiotrophoblast necrosis. Through immunohistochemistry and RNA in situ hybridization, we detected viral antigens and nucleic acid within the syncytiotrophoblast. Although a high number of CD163^+^ Hofbauer cells were observed in the chorionic villi stroma, SARS-CoV-2 was not detected within these macrophages, despite their close proximity to the infected trophoblast [10,11]. This study focused on a transcriptomic analysis comparing SARS-CoV-2-infected placentas to non-infected placentas, as well as the in situ RNA expression of the SARS-CoV-2 and its receptor ACE2 and a set of interleukins and TNF-alpha.

## 2. Results

The Papa Giovanni XXIII Hospital in Bergamo, Italy, has documented three cases of vertical transmission of SARS-CoV-2, occurring at different times. Two cases (Cases 1 and 2) were identified early in the pandemic, between 5 March and 21 April 2020, involving pregnant women in their third trimester who tested positive for COVID-19. The third case, occurring in 2021, involved an asymptomatic woman in her second trimester of a twin pregnancy (Case 3A and B), who delivered two stillborn fetuses due to the premature rupture of membranes. In all cases, the mothers tested positive for COVID-19 via real-time polymerase chain reaction (RT-PCR) from nasopharyngeal (NP) swabs. The placentas of the two women who delivered neonates with SARS-CoV-2 positive NP swabs exhibited chronic histiocytic intervillositis, accompanied by the presence of macrophages in both the intervillous and villous spaces. In the third case, involving the twin pregnancy, SARS-CoV-2 was detected in all tissue samples by RT-PCR, except for the kidney of the second fetus.

### 2.1. Gene Expression Profile

Transcriptomic analysis using NanoString technology (NanoString Technologies, Seattle, WA, USA) revealed significant differences in gene expression between the placentas of mother/newborn pairs affected by COVID-19 (Group 1) and those of the control group (Group 2). In Group 1, according to the Benjamani–Hochberg method [12], 305 genes met the criterion of an adjusted *p*-value of less than 0.05, and 219 genes met the criterion of less than 0.01 (Figure 1). In addition to the high expression levels of SARS-CoV-2-specific genes (Figure 1), Group 1 placentas showed an up-regulation of genes involved in cell–cell signaling (e.g., CCL20, C3, MARCO) and antigen binding and defense response (e.g., LILRA3, CXCL10, CD48, CD86). IL1RN, IL-18, and LILRA3 were also highly expressed in Group 1 placentas, underscoring their role in the inflammatory response to SARS-CoV-2 in affected placentas. We also identified five down-regulated genes (p_adj_ < 0.05), namely MUC1, CR2, AICDA, NOS2, and C9. Table 1 reports the top 25 up-regulated genes with the most significant differential expression between SARS-CoV-2 infected placentae and controls.

### 2.2. Gene Ontology

We included the up-regulated genes (*n* = 300) in Set 1 and the top 25 up-regulated genes with the most significant differential expressions in Set 2 between SARS-CoV-2 affected placentae and controls. GO enrichment analysis was performed only for Set 2, with genes ranked by their significance in differential expression analysis. Table 2 lists GO terms significantly enriched at the top, indicating up-regulated gene expression. These GO terms include biological processes essential for cell killing, inflammatory responses to antigenic stimuli, the positive regulation of TNF family production, leukocyte-mediated immunity, and the positive regulation of cytokine production. Among the top up-regulated genes, C-Type Lectin Domain Containing 7A (CLEC7A) was noted in eight pathways, including cell killing, leukocyte-mediated immunity and positive regulation of TNF family production. Major Histocompatibility Complex, Class II, DR Beta 1 (HLA-DRB1) was involved in seven pathways, including cell killing, inflammatory response to antigenic stimuli and leukocyte-mediated immunity. The complement factor C3 was found in five pathways, including leukocyte-mediated immunity and positive regulation of cytokine production.

### 2.3. Protein–Protein Interactions and Transcription Factor Prediction

Using the online tool STRING, the protein–protein interactions (PPI) among the up-regulated genes (Set 1) and the top 25 up-regulated genes with the most significant differential expressions (Set 2) between SARS-CoV-2-affected placentae and controls were analyzed. For Set 1, 288 nodes and 9083 edges were identified, compared to an expected 1992 edges, resulting in an enrichment score of <1.0 × 10^−16^, indicating higher interaction than expected. For Set 2, 20 nodes and 51 edges were identified, compared to an expected 7 edges, also resulting in an enrichment score of <1.0 × 10^−16^, indicating higher interaction than expected (Figure 2). Identified hub genes, meaning genes connected to other genes in this network, included CXCL10 interacting with eight other proteins, C3, CCL20 and CXCL8 interacting with six other proteins, and CD48 and CLEC7A interacting with five other proteins.

Transcription factor enrichment analysis was conducted to identify which transcription factors were predicted to regulate the top 25 up-regulated genes with the most significant differential expression (Set 2) between SARS-CoV-2-affected placentae and controls (Appendix A).

### 2.4. RNAscope^®^ Image Analysis

RNAscope^®^ technology was used to visualize SARS-CoV-2, its receptor ACE2, and the RNAs of IL1-β, IL-6, IL-8, IL-10 and TNF-α in consecutive sections of FFPE placental tissues. To investigate their distribution in the placenta, a computer-aided image analysis system quantified the RNA-covered surface, expressed as a percentage, in three non-overlapping regions: (a) maternal side (lower third of the placental disk thickness), (b) intermediate zone (middle third) and (c) fetal side (upper third). Table 3 shows the percentages of RNA-covered surfaces in these three regions. Figure 3 shows histological areas with SARS-CoV-2 RNA detected in the fetal side (A), intermediate zone (B) and maternal side (C) of the placenta.

Figure 4A graphically represents the distribution of RNA-covered surfaces in the fetal side, intermediate zone, and maternal side of the placenta for each investigated case. Figure 4B shows the gradients (in gray levels) based on the minimum and maximum percentages of RNA-covered tissue surfaces observed across all four cases.

A non-statistically significant “gradient” for SARS-CoV-2 was observed, with higher surface coverage on the fetal side (2.42 ± 3.71%) compared to the maternal side (0.74 ± 1.19%) of the placenta. Although not statistically significant, the surface area covered by ACE2 mRNA was higher on the maternal side (0.02 ± 0.04%) compared to the fetal side (0.01 ± 0.01%) of the placenta. IL-6 and IL-8 were more prevalent on the fetal side (0.03 ± 0.04% and 0.06 ± 0.08%, respectively) compared to the maternal side (0.02 ± 0.01% and 0.02 ± 0.02%, respectively). The mean surface areas of IL-1β and IL-10 were found to be equal on both the fetal (0.04 ± 0.04% and 0.01 ± 0.01%, respectively) and maternal sides of the placenta (0.04 ± 0.05% and 0.01 ± 0.01%, respectively). The mean surface area of TNF-α was found to be equal on both the fetal and maternal sides of the placenta (0.02 ± 0.02% and 0.02 ± 0.02%, respectively). On the maternal side, ACE-2 and all examined interleukins, but not TNF-α, exhibited an inverse mRNA amount compared to SARS-CoV-2. On the fetal side, ACE-2, IL-6 and IL-8 were inversely correlated with SARS-CoV-2 (r = −0.3, r = −0.1 and r = −0.4, respectively), while IL-1β and IL-10 showed positive correlations (r = 0.9, *p* = 0.005 and r = 0.5, respectively). TNF-α exhibited a positive correlation with SARS-CoV-2 on both maternal (r = 0.4) and fetal sides (r = 0.9) of the placenta.

## 3. Discussion

Pregnant women and their unborn or newborn babies are considered a vulnerable demographic when it comes to developing severe COVID-19. Since the start of 2020, there have been varying reports worldwide regarding the possibility of severe acute respiratory syndrome coronavirus-2 (SARS-CoV-2) being transmitted vertically and its impact on maternal health [13,14,15]. Moreover, pregnant populations belonging to low- and middle-income countries (LMICs) have been reported to have higher miscarriages and stillbirths as well as neonatal SARS-CoV-2 infections [16]. The current study focused on a transcriptomic analysis aimed at examining differences in gene expression between placentas from mother–newborn pairs affected by COVID-19 and those from unaffected controls. Additionally, it investigated the in situ expression of molecules involved in placental inflammation, providing insights into how the coronavirus potentially traverses the maternal–fetal interface. We found significant differences in gene expression between the placentae of COVID-19-affected mother/newborn pairs and controls. Affected placentae showed significant changes in 305 genes (p_adj_ < 0.05) or 219 genes (p_adj_ < 0.01). Genes involved in cell signaling (e.g., CCL20, C3, MARCO), immune response (e.g., LILRA3, CXCL10, CD48, CD86) and the high expression of IL1RN, IL-8 and LILRA3 in Group 1 suggest their role in the inflammatory response to SARS-CoV-2. It has been reported that the SARS-CoV-2 virus activates maternal and placental immune responses [17].

It has been shown that SARS-CoV-2 can damage human placentas, leading to pregnancy complications such as preeclampsia and premature birth [18]. Notably, among the top 25 up-regulated genes, we identified the complement factor C3. Recently, Blakey et al. [19] confirmed that excessive placental complement deposition is associated with significant concurrent changes in maternal and fetal circulating complement biomarkers in preeclampsia. Our analysis also identified five down-regulated genes (p_adj_ < 0.05): MUC1, CR2, AICDA, NOS2 and C9. Among these, AICDA, C9 and CR2 were involved in multiple pathways, primarily related to complement activation, leukocyte-mediated immunity and the adaptive immune response.

With RNAscope technology, combined with image analysis, we detected varying percentages of RNA-covered surfaces between the maternal and fetal sides of the placenta (Figure 3 and Figure 4). Specifically, SARS-CoV-2 RNA and its receptor ACE2 were detected in both sides of the placenta in all cases investigated, as previously demonstrated by other studies [20,21]. Figure 3 shows Case 1, where we found a higher presence SARS-CoV-2 RNA on the fetal side of the placenta. The expression of the receptor ACE2 exhibited variability without a consistent pattern across cases. Notably, the twins displayed comparable RNA-covered surfaces in the intermediate zone and fetal side of the placenta. In contrast, Case 1 and Case 2, while comparable to each other, exhibited a gradient opposite in the fetal side of the placenta compared to that observed in the twins. All examined interleukins demonstrated an inverse correlation with SARS-CoV-2 RNA, except for IL-1β (r = 0.9) and IL-10 (r = 0.5) on the fetal side of the placenta. Conversely, TNF-α showed a direct correlation with SARS-CoV-2 RNA on both the fetal and maternal sides of the placenta, with correlation coefficients of r = 0.9 and r = 0.4, respectively. The study also revealed significant heterogeneity in SARS-CoV-2 RNA, ACE2 RNA and the RNA of IL1-β, IL-6, IL-8, IL-10 and TNF-α-covered surfaces among the cases, with pronounced differences observed, particularly in the twins. The above results confirm our previous studies that first reported cases with positive PCR results for SARS-CoV-2 in the mother, neonate and the placental tissues. RNAscope^®^ technology allows us to visualize SARS-CoV-2, its spike protein receptor ACE2 RNA and a set of interleukins (IL1-β, IL-6, IL-8, IL-10), as well as TNF-α, an inflammatory cytokine produced by macrophages and monocytes during acute inflammation, responsible for various cellular signaling events leading to necrosis or apoptosis. All these targets were measured using a computer-aided image analysis system, assessing their covering area. This combined method retains tissue morphology, a feature lost with molecular-based methods such as PCR. For all the targets, we found differences in the RNA-covered surface across the two-dimensional tissue sections, which persisted when studying a set of sub-regions (Figure 5). This non-uniform distribution of RNA probes underscores the variability observed in the study. A notable finding in the twins was the varying amounts of RNA-covered surfaces across all probes, with one twin showing the lowest amounts compared to all the other cases investigated (Figure 4B). Morales et al. [22] reported a case involving a 28-year-old non-obese woman with a two-year history of primary infertility who underwent in vitro fertilization (IVF). After endometrial preparation, two chromosomally normal embryos, one female and one male, were transferred. Twenty days post-embryo transfer, two gestational sacs were identified, and at 6.3 weeks, the vitality of both embryos was confirmed. At 30 weeks, the patient developed flu-like symptoms, including rhinorrhea, mild nasal congestion, otalgia, odynophagia and a productive cough, but without fever or respiratory distress. She tested positive for COVID-19 and subsequently delivered a boy and a girl. RT-PCR testing for SARS-CoV-2 was conducted on the newborns; the male tested negative, while the female tested positive. Additionally, the female placenta tested positive for SARS-CoV-2, whereas the male placenta tested negative.

In our case, we observed that one twin had a greater SARS-CoV-2 RNA-covered surface on the maternal side and intermediate zone of the placenta, while the other twin exhibited a greater RNA-covered surface on the fetal side and intermediate zone. When comparing the SARS-CoV-2 RNA-covered surfaces of all investigated subjects, both twins showed lower RNA-covered surfaces overall. Additionally, it has been reported that vertical transmission is possible due to the presence of ACE2 receptors in the uterus and placenta [23]. Similarly, in the twins, we found that the ACE2 RNA-covered surface followed the same pattern as the SARS-CoV-2 RNA-covered surface. The presence of both ACE2 and SARS-CoV-2 RNA was also observed in the other two cases studied.

Maternal COVID-19 status was categorized as mild, severe or critical [24]. As reported by Penfield et al., of 11 placental or membrane swabs sent for testing after delivery, 3 swabs returned with positive results for SARS-CoV-2, all in women with severe to critical COVID-19 at time of delivery [25]. In another study, Mehta et al. [26] reported a case of a 39-year-old woman who had conceived twins via IVF, who presented at 27 weeks of gestation with nasal congestion and dry cough for 7 days. Her PCR resulted positive for SARS-COV2. One of the twins tested positive for SARS-CoV2, associating this finding with a vertical transmission.

It is known that pregnancy is a unique immunologic state in which the maternal immune system is modulated to allow tolerance to paternally derived fetal antigens, thereby leading to an increased susceptibility to infections, especially pneumonia [27]. In one study, up to 25% of pregnant women with pneumonia required critical care services, including mechanical ventilation [28]. The importance of the maternal immunological state in COVID-19 has been highlighted by Schwartz and colleagues [23,24], who proposed that SARS-CoV-2 placentitis, placental destruction and stillbirth may result from a failure of maternal–fetal tolerance and the rejection of such fetal-derived tissues as the placenta, analogous to the solid organ rejection syndromes that can occur following allogeneic transplantation.

The above considerations regarding SARS-CoV-2 infection during pregnancy, immune status and vertical transmission observed in one of the twins may explain the heterogeneity noted in our study. This heterogeneity relates to the gradients of SARS-CoV-2, its receptor ACE2, immunological interleukins and the inflammatory cytokine TNF-α between the maternal and fetal sides of the placenta across different cases and within the twin pairs.

Research is limited given that infant infection is rare, approximately 4% as quoted in one systematic review [29], and screening for fetal infection at the time of birth is not a current standard of care. It is important to note that infant infection includes postnatal infection and, in fact, none of 28 neonates who tested positive met criteria for intrauterine infection criteria described by Shah et al. [30]. It has been shown that due to the recent nature of the disease, few studies are found in the literature about the vertical transmission of SARS-CoV-2 [31,32]. In all case reports and case series, the mothers’ infection occurred in the third trimester of pregnancy, there were no maternal deaths, and most neonates had a favorable clinical course [33].

It is true that RNA and protein expression profiling and other technologies have revolutionized how clinicians study the molecular basis of human pathologies and drug effects. These technologies promise efficient, high-throughput methods to delineate mechanisms of action and to predict disease progression. However, achieving this requires understanding methodological constraints to design genome-scale studies and interpret large data sets. The complexity of biological processes and technical variability such as sample preparation, RNA quality and platform differences continue to pose significant challenges. In conclusion, despite being based on only four cases, this study sheds light on the immunological molecules involved in placental inflammation. This approach represents a further step in elucidating the potential mechanism by which coronavirus crosses the maternal–fetal interface.

## 4. Materials and Methods

### 4.1. Patients

Case 1. The mother, 27 years old, who was symptomatic with fever and cough, tested positive for COVID-19 at 37.6 weeks of gestation. She subsequently entered spontaneous labor and delivered a male neonate weighing 2660 g via vaginal delivery. The Apgar scores were 9 and 10 at 1st and 5th min, respectively, and the umbilical artery pH was 7.28. The newborn tested positive for COVID-19 from nasopharyngeal (NP) swabs obtained immediately after birth, 24 h later and after 7 days. Despite testing positive, the neonate remained asymptomatic, except for mild initial feeding difficulties. After 10 days of observation, the newborn was discharged without complications.

Case 2. The mother, 35 years old, symptomatic with fever and cough, tested positive for COVID-19 at 35.1 weeks of gestation. The newborn was delivered by cesarean section due to non-reassuring fetal status. The female neonate, weighing 2686 g, had Apgar scores of 9 and 10 at 1st and 5th min, respectively, and an umbilical artery pH of 7.32. Due to her prematurity, she was transferred to the neonatal intensive care unit immediately after birth. In contrast to the previous case, the first nasopharyngeal (NP) swab conducted immediately after birth yielded a negative result; however, a follow-up test on day 7 returned a positive for SARS-CoV-2. It is important to note that during this period, the newborn did not have any contact with the mother and was isolated from other neonates. After hospitalization for routine late preterm care, she was discharged on day 20 of life.

Case 3. A 35-year-old asymptomatic woman with a twin pregnancy achieved through in vitro fertilization (IVF) presented at 20 weeks of gestation to the gynecology and obstetrics emergency room due to preterm premature rupture of the membranes in a high-risk pregnancy. She had previously undergone cervical cerclage at 16 weeks for cervical incompetence. Upon entrance screening, she tested positive for COVID-19. By 21.4 weeks, her clinical condition deteriorated, manifesting with fever, increased leukocyte count and elevated PCR value. Premature labor ensued, necessitating the removal of cervical cerclage, following which the patient delivered two stillborn fetuses vaginally.

### 4.2. Hematoxylin and Eosin Staining

Hematoxylin and eosin (H&E) staining of 4 µm thick formalin-fixed paraffin-embedded (FFPE) tissue sections was conducted using the DAKO CoverStainer (DAKO, Glostrup, Denmark). The instrument utilized DAKO Ready-to-Use (RTU) reagents and validated pre-optimized protocols. Images were digitized using an Axio Zeiss Scope A1 (Zeiss, Oberkochen, Germany) microscope.

### 4.3. Immunohistochemical Staining

The 4 µm thick formalin-fixed paraffin-embedded (FFPE) slides underwent the following procedures: They were deparaffinized and rehydrated using xylene followed by absolute ethanol, and then washed with tap water. Endogenous peroxidase activity was blocked by incubating slides in methanol containing 10% H_2_O_2_ for 15 min. Antigen retrieval was achieved by heating the slides in Retrieve All 2 buffer (pH 10.0) (Covance) in a steamer for 20 min. To prevent nonspecific staining, slides were incubated with an UltraVision protein block (Thermo-Scientific, Waltham, MA, USA) for 8 min. Next, the slides were incubated overnight at 4 °C with a mouse monoclonal primary antibody against the SARS-CoV-2 viral nucleocapsid protein (NC, 1:100, Sino Biological Inc., Beijing, China) diluted in PBS with 1% bovine serum albumin (BSA, Sigma-Aldrich, St. Louis, MI, USA). After rinsing with phosphate-buffered saline (PBS) containing 0.05% Tween20 (PBS-T, both from Sigma), slides were incubated for 30 min at room temperature with a goat anti-mouse horseradish peroxidase (HRP)-conjugated secondary antibody (EnVision, DAKO). Specimens were developed using 3,3-diaminobenzidine tetrahydrochloride (DAB substrate kit, AbCAM, Cambridge, UK), counterstained with Gill’s Hematoxylin No. 2 (Sigma), and mounted with EuKitt (Bio-Optica, Milano, Italy). Negative controls for nucleocapsid protein included slides from SARS-CoV-2-uninfected placentas (*n* = 2). Images were captured using an Eclipse E800 microscope equipped with a cooled digital camera (DS-U1) and analyzed using LuciaG 5.0 software (Nikon, Tokyo, Japan).

### 4.4. RT-PCR Frozen Samples

We collected a sample of each placental biopsy in Universal Transport Medium (Copan Italia, Brescia, Italy) and stored them at −80 °C in Biobank after treatment with RNAlater-ICE (ThermoFisher Scientific, Waltham, MA, USA). Later on, a small piece of placenta (about 4–5 mm^3^) was processed with 50 μL of proteinase K (QIAGEN, Hilden, Germany) and 200 μL of Tris-EDTA buffer solution (Sigma-Aldrich, Darmstadt, Germany) for an hour.

In order to detect the presence of the virus in the samples, an RT-PCR was performed using 200 μL of digested sample with a SARS-CoV-2 ELITe MGB kit in association with Elite InGenius platform (ELITech Group, Puteaux, France) in the Microbiology and Virology Laboratory. This method targets two specific regions of the viral genome: the RdRp (RNA-dependent RNA polymerase) gene and the ORF8 (Open reading frame) gene that encoded an accessory viral protein.

### 4.5. RT-PCR FPPE Samples

For each FFPE specimen, two 10 µm thick tissue sections were collected, and all paraffin was removed by treating them with deparaffinization solution (QIAGEN). RNA was extracted according to the instructions of the RNeasy FFPE Kit (QIAGEN) and the quantity was determined with a nanodrop spectrophotometer. The RNA extracted was assayed for the qualitative detection of Novel Coronavirus (2019-nCoV) with a real-time reverse transcription (RT)-PCR) (AmoyDx Novel Coronavirus (2019-nCoV) detection kit according to manufactory protocol. The reverse transcription of RNA was extracted and PCR amplification was performed in a one-step procedure. Briefly, the kit was designed for a specific amplification of the ORF1ab (open reading frame, ORF 1ab) and N (Nucleoprotein, N) conserved regions of novel coronavirus SARS-CoV-2 in viral RNA. The targeted region was amplified by specific primers and detected by fluorescence probes. A non- competitive internal control was included in the nCoV RNA detection system to assess RNA quality and monitor the whole PCR procedure.

### 4.6. RNA In Situ Hybridization

The single-molecule RNA in situ hybridization of SARS-CoV-2 RNA was detected by using RNAscope^®^ technology (Advanced Cell Diagnostics, Newark, CA, USA), an RNA in situ hybridization (ISH) technique. Paired double-Z oligonucleotide probes were designed for hybridization to the target RNA by using custom software. The RNAscope 2.5 LS Probe V-nCoV2019-S (catalog number 848568; Advanced Cell Diagnostics, Newark, CA, USA) was used. The RNAscope^®^ 2.5 LSx Reagent Kit-Brown (Advanced Cell Diagnostics) in combination with a BOND-III automated stainer (Leica Biosystems, Buffalo Grove, IL, USA) was used to process the samples according to the manufacturer’s instructions. The FFPE tissue section samples were prepared according to manufacturer’s recommendations. The RNA integrity of each sample was evaluated with a probe designed for hybridization specifically to the ubiquitin C and cyclophilin B housekeeping genes. The negative control background staining was evaluated using a probe specific to the bacterial dapB gene. Each punctate dot signal representing a single target RNA molecule could be detected with standard light microscopic analysis. Double staining immunohistochemistry was carried out for CD163/RNA in situ hybridization for SARS-CoV-2. Immunohistochemical staining with NovocastraTM Liquid Mouse Monoclonal Antibody CD163 (10D6 clone) was used to identify Hofbauer cells in placenta combined with RNAscope^®^ 2.5 LS Probe V-nCoV2019-S. The double staining was performed on a BOND-III Automated stainer (Leica Biosystems, Buffalo Grove, IL, USA) according to the manufacturer’s instructions.

### 4.7. RNAscope^®^ Image Analysis

Tissue sections were digitized at 20× magnification (0.22 µm pixel width and height) using ZEISS Axioscan.Z1 (ZEISS, Oberkochen, Germany). Images were analyzed using an ISH Tissue Analyzer (Histology Core, IRCCS Humanitas Research Hospital, Rozzano, Milan, Italy). For each slide, three ROIs (90 mm^2^ each) representing the maternal and fetal sides of the placenta plus another intermediate region were selected. The gradient (G) of expression between the maternal and fetal sides of the placenta, was obtained using Formula (1),
(1)$$G=255-\left(\frac{x-min}{max-min}\right)*255$$
which determines the gray intensity level proportionate to the percentage of RNA-covered surface in the maternal, intermediate and fetal zones for each marker. In a grayscale range from 0 (absolute black color) to 255 (absolute white color), ∗ represents the percentage of RNA-covered surface, while min and max represent the minimum and maximum percentages of RNA-covered surfaces among the three zones, respectively.

### 4.8. mRNA Expression according to Nanostring

A small piece of placenta (from a central cotyledon, comprising both fetal and maternal sides) was collected at delivery, washed in PBS and placed in “RNA-later” medium for a maximum of one week. Specimens were FFPE-fixed, according to national procedures for the manipulation of SARS-CoV-2-positive samples. We collected placental samples of (1) mothers positive for SARS-CoV-2 infection, with SARS-CoV-2 positive (+/+) newborns; (2) mothers positive for SARS-CoV-2 infection with SARS-CoV-2 negative (+/−) newborns; and mother/newborn pairs negative for SARS-CoV-2 infection, as controls (−/−). These latter were collected and treated with the same protocol as that of the COVID^+^ placentae. RNA was extracted from the FFPE tissue with the High Pure FFPET RNA Isolation Kit (Roche Diagnostics, Monza, Italy), following the manufacturer’s instructions. The RNA expression of about 500 genes involved in immune response was investigated by the Nanostring Platform by the nCounter Human Immunology kit, integrated with a pool of probes specific for the identification of 8 SARS-CoV-2 genes and 2 ACE2 receptor genes. This panel included major classes of cytokines and their receptors, enzymes with specific gene families such as the major chemokine ligands and receptors, interferons and their receptors, the TNF-receptor superfamily and the KIR family genes. Additionally, 100 ng of total RNA was used as the input into hybridization reactions containing reporter and capture probes, according to the manufacturer’s instructions. RNA samples were hybridized at 65 °C for 24 h. Post-hybridization processing was carried out in an nCounter MAX System. RCC files were compiled and analyzed using nSolver analysis software (version 4.0), as per the instructions of the manufacturer. The internal reference genes included in the probe set were used for normalization, following the manufacturer’s guidelines.

### 4.9. Statistical Analysis

The data were expressed as mean ± standard deviation. Statistical differences among variables were analyzed by an unpaired two-tailed *t*-test. Correlations between variables were quantified using Pearson’s correlation coefficients (rho). The coefficient of variability was calculated by dividing the standard deviation by the mean value and then expressing the result as a percentage. *p* < 0.05 was considered to be statistically significant.

### 4.10. Bioinformatic Analysis of Differentially Expressed Genes

Gene filtering and data transformation were performed, excluding genes with very low expression. Differential expression analyses were conducted using nSolver, and *p*-values were adjusted using the Benjamini–Hochberg method [12]. Genes were considered statistically significant if their adjusted *p*-values were lower than 0.05. The visualization of these differentially expressed genes was achieved through the construction of volcano plots using Python (version 3.11.2). Volcano plots juxtapose the significance of each gene’s differential expression (*p*-value) against the magnitude of its change (fold change).

A Gene Ontology (GO) enrichment analysis was performed using GO annotation from the org.Hs.eg.db Bioconductor package version 3.19, with mapping between GO identifiers and GO terms taken from GO.db version 3.19. Biological processes among all GO terms were considered. Enrichment test *p*-values were also adjusted using the Benjamini–Hochberg method [12]. The Q-value (the minimum false discovery rate at which an observed score is considered significant) was assessed. All analyses were conducted in R version 4.4.1. Gene lists associated with each GO term were scanned for the top 25 up-regulated genes identified by differential gene expression analysis.

### 4.11. Protein–Protein Interaction Analysis and Transcription Factor Prediction

We utilized the Search Tool for the Retrieval of Interacting Genes/Proteins (STRING) to explore functional protein association networks (string-db.org) [34]. For the top 25 up-regulated genes, we set the following parameters: organism, Homo sapiens; meaning of network edges, evidence; active interaction sources, text mining, experiments, databases, co-expression, gene fusion and co-occurrence. We set the minimum required interaction score to a low confidence of 0.4 and placed no limitations on the number of interactions. Additionally, the ChEA3 database was used to identify transcription factors related to the differentially expressed genes. ChEA3, a transcription factor enrichment tool, ranks transcription factors within user-submitted gene sets using the mean rank method. The target genes of selected transcription factors were further investigated using information from GeneCards https://genecards.org (accessed on 26 June 2024).

## Figures and Tables

**Figure 1 ijms-25-08825-f001:**
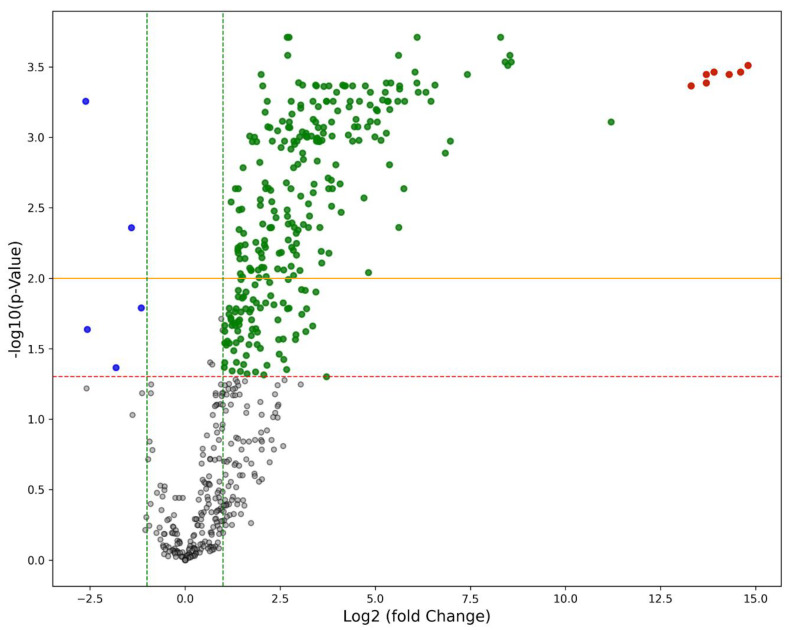
The volcano plot illustrates differential gene expression between SARS-CoV-2-infected placentae and control placentae. Up-regulated genes are shown as green dots, down-regulated genes as blue dots, and SARS-CoV-2-specific genes as red dots. Grey dots represent statistically non-significant genes. Thresholds are set at −log10 adjusted *p*-values greater than 1.3 (red dashed line) and 2 (yellow line), corresponding to statistical significance levels of 0.05 and 0.01, respectively, with|log2 fold changes|greater than 1 (green dashed line).

**Figure 2 ijms-25-08825-f002:**
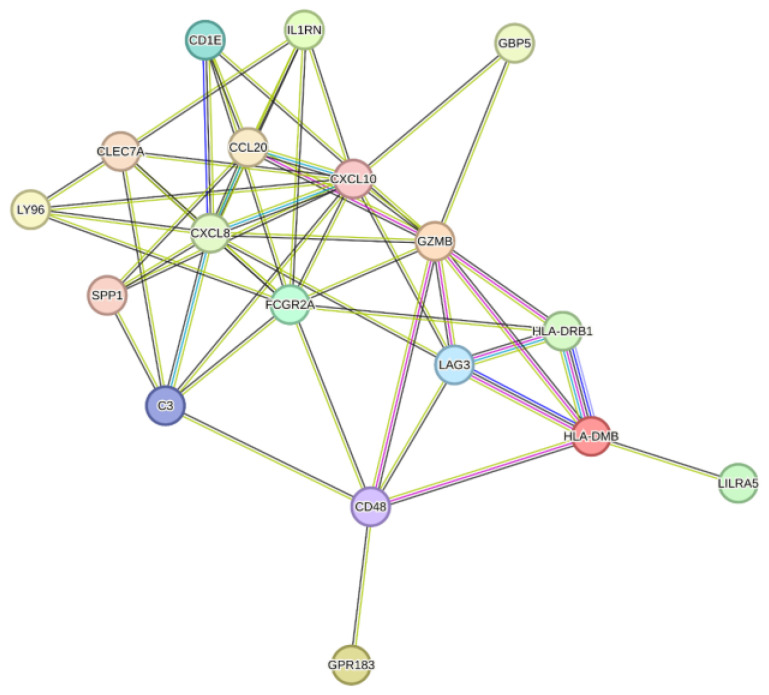
Analysis of the top 25 up-regulated genes with the most significant differential expressions (Set 2) between SARS-CoV-2-affected placentae and controls analyzed through the STRING database.

**Figure 3 ijms-25-08825-f003:**
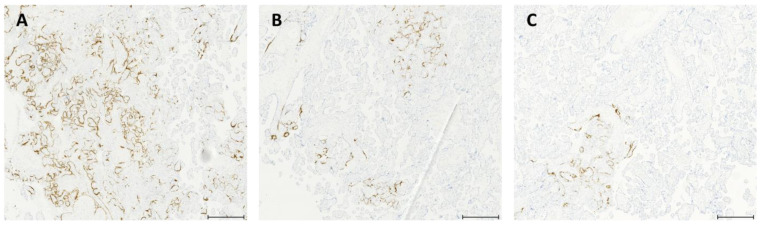
Placental tissue showing SARS-CoV-2 RNA distribution in fetal side (upper third) (**A**), intermediate zone (middle third) (**B**) and maternal side (lower third of the placental disk thickness) (**C**). In all images, the positive signal is confined to the syncytiotrophoblast of the chorionic villi. Scale bar: 500 μm.

**Figure 4 ijms-25-08825-f004:**
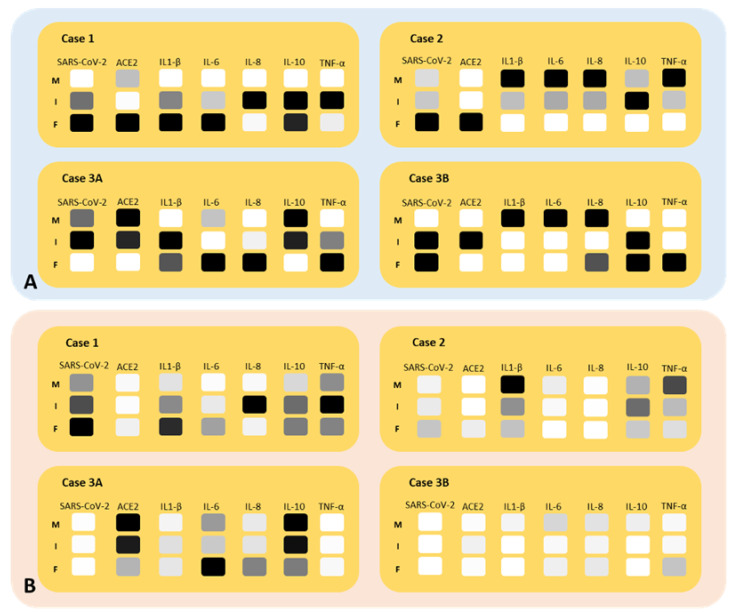
(**A**) Visual representations of “gradients” from the maternal side, through the intermediate zone, to the fetal side of the placenta. The color black indicates a “high” RNA-covered surface for the specific probe, while lighter shades of gray indicate progressively “lower” RNA-covered surfaces. (**B**) Gradients based on the minimum and maximum RNA-covered surfaces of single probes observed across all four cases.

**Figure 5 ijms-25-08825-f005:**
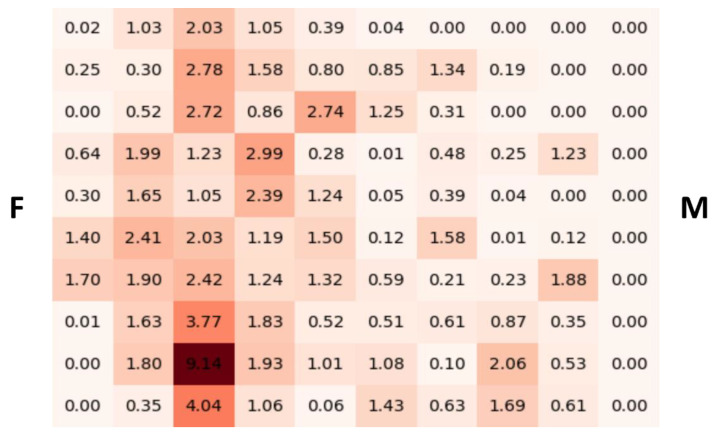
A heatmap showing the two-dimensional distribution of SARS-CoV-2 RNA, displaying different amounts from the fetal zone (F) of the placenta towards the maternal side (M) in Case 1. The values within the subregions represent the percentage of mRNA-covered surface. SARS-CoV-2 RNA was found to be heterogeneously distributed with a coefficient of variability of 128%.

**Table 1 ijms-25-08825-t001:** Top 25 up-regulated genes with the most significant differential expression between SARS-CoV-2-infected placentae and controls.

Gene Title	Gene Symbol	Log2 Fold-Change	95% CI	*p*-Value	Padj
G Protein-Coupled Receptor 183	GPR183	2.68	[2.08–3.27]	1.34 × 10^−6^	0.0002
Complement C3	C3	2.73	[2.14–3.33]	1.11 × 10^−6^	0.0002
Major Histocompatibility Complex, Class II, DR Beta 1	HLA-DRB1	6.1	[4.76–7.44]	1.21 × 10^−6^	0.0002
Guanylate Binding Protein 5	GBP5	8.29	[6.49–10.1]	1.04 × 10^−6^	0.0002
Lymphocyte Antigen 96	LY96	2.7	[2.05–3.34]	3.14 × 10^−6^	0.00026
Lymphocyte Activating 3	LAG3	5.61	[4.27–6.94]	2.77 × 10^−6^	0.00026
C-C Motif Chemokine Ligand 20	CCL20	8.54	[6.51–10.6]	2.70 × 10^−6^	0.00026
C-X-C Motif Chemokine Ligand 10	CXCL10	8.41	[6.31–10.5]	4.52 × 10^−6^	0.00029
Leukocyte Immunoglobulin Like Receptor A3	LILRA3	8.57	[6.44–10.7]	4.28 × 10^−6^	0.00029
Interleukin 1 Receptor Antagonist	IL1RN	8.48	[6.31–10.7]	5.83 × 10^−6^	0.00031
ORF8 protein	ORF8	14.8	[11.1–18.6]	5.42 × 10^−6^	0.00031
Interleukin 8	IL8	6.04	[4.45–7.63]	7.75 × 10^−6^	0.00034
Nucleocapsid phosphoprotein	Nucleocapsid phosphoprotein	13.9	[10.3–17.6]	7.91 × 10^−6^	0.00034
Surface glycoprotein	Surface glycoprotein	14.6	[10.7–18.4]	8.29 × 10^−6^	0.00034
Major Histocompatibility Complex, Class II, DM Beta	HLA-DMB	2	[1.46–2.55]	1.11 × 10^−5^	0.00036
Leukocyte Immunoglobulin Like Receptor A5	LILRA5	7.42	[5.41–9.43]	1.03 × 10^−5^	0.00036
ORF7a protein	ORF7a	13.7	[9.93–17.4]	1.11 × 10^−5^	0.00036
Membrane glycoprotein	Membrane glycoprotein	14.3	[10.4–18.1]	1.02 × 10^−5^	0.00036
C-Type Lectin Domain Containing 7A	CLEC7A	2.98	[2.16–3.81]	1.34 × 10^−5^	0.00041
Granzyme B	GZMB	5.29	[3.86–6.72]	1.62 × 10^−5^	0.00041
CD48 Molecule	CD48	5.41	[3.89–6.92]	1.44 × 10^−5^	0.00041
Macrophage Receptor with Collagenous Structure	MARCO	6.09	[4.37–7.82]	1.56 × 10^−5^	0.00041
ORF1ab Polyprotein	orf1ab	13.7	[9.87–17.6]	1.55 × 10^−5^	0.00041
Secreted Phosphoprotein 1	SPP1	3.09	[2.19–3.98]	1.98 × 10^−5^	0.00043
Fc Gamma Receptor IIa	FCGR2A	3.43	[2.45–4.42]	1.82 × 10^−5^	0.00043

**Table 2 ijms-25-08825-t002:** Gene ontology of the top 25 up-regulated genes with the most significant differential expression between SARS-CoV-2-affected placentae and controls.

Pathway	Stat. Mean	Stat. sd	*p*. Val	q. Val	Set. Size
GO:0001906 cell killing	4.92	2.01	9.10 × 10^−10^	2.91 × 10^−7^	226
GO:0002437 inflammatory response to antigenic stimulus	5.19	2.63	1.06 × 10^−6^	1.59 × 10^−4^	84
GO:0098581 detection of external biotic stimulus	3.93	1.89	1.86 × 10^−6^	1.59 × 10^−4^	26
GO:0032760 positive regulation of tumor necrosis factor production	4.13	2.21	3.02 × 10^−6^	1.59 × 10^−4^	109
GO:0002443 leukocyte-mediated immunity	4.36	1.48	3.27 × 10^−6^	1.59 × 10^−4^	469
GO:0002699 positive regulation of immune effector process	3.88	1.84	3.33 × 10^−6^	1.59 × 10^−4^	257
GO:1903557 positive regulation of tumor necrosis factor superfamily cytokine production	4.13	2.21	3.48 × 10^−6^	1.59 × 10^−4^	113
GO:0001819 positive regulation of cytokine production	4.59	2.56	4.73 × 10^−6^	1.89 × 10^−4^	500
GO:0009595 detection of biotic stimulus	3.93	1.89	7.02 × 10^−6^	2.50 × 10^−4^	40
GO:0001909 leukocyte-mediated cytotoxicity	5.11	1.17	9.12 × 10^−6^	2.92 × 10^−4^	144

**Table 3 ijms-25-08825-t003:** The percentage of RNA-covered surfaces of SARS-CoV-2, ACE2, IL1-β, IL-6, IL-8, IL-10 and TNF-α measured through a computer-aided image analysis system on placenta tissues. Two cases (Cases 1 and 2) involved pregnant women in their third trimester who tested positive for COVID-19. The third case involved an asymptomatic woman in her second trimester of a twin pregnancy (Case 3A and 3B) who delivered two stillborn fetuses due to the premature rupture of membranes.

	Placenta Zone	Case 1	Case 2	Case 3	
				A	B
SARS-CoV-2	M	2.5	0.384	0.057	0.001
	I	5.489	0.639	0.08	0.004
	F	7.845	1.785	0.026	0.004
ACE2	M	0.004	0.002	0.087	0.003
	I	0.003	0.002	0.078	0.007
	F	0.007	0.008	0.027	0.003
IL1-β	M	0.017	0.117	0.009	0.009
	I	0.056	0.053	0.018	0.004
	F	0.098	0.031	0.015	0.004
IL6	M	0.006	0.011	0.037	0.018
	I	0.012	0.007	0.022	0.007
	F	0.035	0.005	0.086	0.01
IL8	M	0.013	0.007	0.035	0.044
	I	0.359	0.005	0.043	0.019
	F	0.022	0.004	0.178	0.036
IL10	M	0.006	0.011	0.034	0.003
	I	0.02	0.02	0.032	0.001
	F	0.018	0.008	0.018	0.001
TNF-α	M	0.035	0.056	0	0.002
	I	0.078	0.021	0.001	0.002
	F	0.038	0.01	0.002	0.018

Notes: The placental regions are designated as the maternal side (M), the intermediate zone (I) and the fetal side (F).

## Data Availability

Data are contained within the article and Appendix A.

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
