# Peer review of "In Situ Analyses of Placental Inflammatory Response to SARS-CoV-2 Infection in Cases of Mother–Fetus Vertical Transmission"

_ijms, 2024, doi:10.3390/ijms25168825_

Round 1

Reviewer 1 Report

Comments and Suggestions for Authors

Manuscript Title:  In Situ Analyses of Placental Inflammatory Response to 2 SARS-CoV-2 Infection in Cases of Mother–Fetus Vertical Transmission

Authors: Morotti D et al

Abstract:  Transcriptomics analysis revealed significant differences in gene expression between the placentae of COVID-19-affected mother/newborn pairs and controls. Affected placentae showed significant changes in 334 genes (p <0.05) or 334 genes (p<0.01).  A reproducible statistical analysis plan should be included in the Methods section to support these findings and other findings reported in the abstract.

There appear to be multiple potential sources of correlation in the data (placentas from mother-newborn pairs; fetal and maternal sides of the placenta; fetal side, intermediate zone and maternal side of the placenta; regions of interest; and twins).  This potential lack of independence in the data is important to address through inference or hypothesis testing.  The Benjamini-Yekutieli false discovery rate method accounts for this expectation that significant differences or changes in genes may be correlated with or dependent on each other.

RNAscopeâ 31 technology, combined with image analysis, revealed differing mRNA levels of ACE2, SARS-CoV-2, 32 IL1-β, IL-6, IL-8, IL-10, and TNF-α between the maternal and fetal sides of the placenta. Please address the statistical methods used for these comparisons. Bioinformatic analysis of differentially expressed genes should also be addressed.  Limitations of gene expression analysis might include technical variability that can affect the reproducibility of results.

Furthermore, comparing gene expression between the fetal and maternal sides of the placenta revealed that all differentially expressed genes were overexpressed on the fetal side. How were these changes in gene expression summarized and analyzed across samples (fold-change estimates, ratios).  Discuss replicates and appropriate statistical tests and multiple testing correction.

In the following sections, we discuss creating ratios and fold-change estimates, and for those experiments with sufficient biological replicates, we discuss appropriate statistical tests and multiple testing correction.

Figure 2 shows SARS-CoV-2 RNA in fetal side, intermediate zone and maternal side of the placenta. Figure 3A graphically represents the distribu- 119 tion of RNA-covered surfaces in the fetal side, intermediate zone, and maternal side of the 120 placenta for each investigated case. Figure 3B illustrates the gradients based on the mini- 121 mum and maximum amounts of single RNA probes observed across all investigated 122 cases. With the exception of a twin (Case 3A),

For each slide, three ROIs were taken at the same size (90mm2 each). 356 These ROIs represented maternal and fetal side of the placenta plus another intermediate 357 region. To calculate the gradient (G) of expression between the maternal and fetal sides of 358 the placenta, we utilized the following formula,

transcriptomics analysis aimed at examining differences in gene expression between placentas from mother-newborn pairs affected by COVID-19 and those from unaffected controls.

Additionally, it investigates the in-situ expression of molecules involved in placental inflammation, providing insights into how the coronavirus potentially traverses the maternal-fetal interface.

A heatmap showing the spatial distribution of COVID mRNA, displaying a gradient from 154 the fetal zone (F) of the placenta towards the maternal side (M) in Case 1. The values within the 155 subregions represent the percentage of mRNA-covered surface

Additionally, genes involved in cell signaling (e.g., CCL20, C3, MARCO) and immune response (e.g., 172LILRA3, CXCL10, CD48, CD86) were upregulated in group 1. High expression of IL1RN, IL18, and LILAR3 in group 1 suggests their role in the inflammatory response to SARS-CoV-2.  With RNAscope technology, combined with image analysis, we observed varying percentages of RNA-covered surfaces between the maternal and fetal sides of the placenta (Figure 2 and 3). Specifically, SARS-CoV-2 RNA showed a gradient with higher expression on the fetal side of the placenta (Figure 3A).

Bioinformatic analysis of differentially expressed genes

Gene filtering and data transformation was performed. Genes with very low expression were excluded (less than 3 samples with at least 10 counts). Gene expression data quality was assessed using variance mean dependency plots, heatmap of euclidean distances between samples and centered principal component analyses. All differential expression analysis was performed on the raw count data from these 20 samples afer filtering out low expression genes. DESeq2 version 1.32.0 was used for differential expression analyses.

Given that a filtering based on low expression was already performed, no additional independent fltering was done in DESeq2. P-values were adjusted by using the Benjamini–Hochberg method21 and genes were considered statistically significant, if adjusted p-values was lower than 0.05. Te frst diferential expression was paired by mother 3 Vol.:(0123456789) Scientifc Reports | (2024) 14:4210 | https://doi.org/10.1038/s41598-024-54724-3 www.nature.com/scientificreports/ identifer with no other adjustment variables in model 1. Te analysis was further adjusted by mode of delivery (categorized as spontaneous delivery vs. instrumental delivery which included deliveries with vacuum extraction, cesarean section or emergency cesarean-section.) in model 222. Likelihood ratio tests were performed to assess whether interaction terms should be added in the models. Tese likelihood ratio tests compared models with and without interaction terms and 0.053% of the genes (8 genes) had adjusted p-values below 0.05 (p-value distribution close to uniform) so it was decided to not have interaction terms in the models. Te last model 3 was adjusted for delivery mode, maternal BMI and gestational age at birth provided as continuous variables in days22. No likelihood ratio tests were performed on these fully adjusted models. For top signifcant genes of each analysis, normalized counts (variance stabilizing transformed data) were plotted for visualizing results. A Gene Ontology (GO) enrichment analysis was performed on the unadjusted model 1 and fully adjusted model 3 analysis adjusting for delivery mode, maternal BMI, and gestational age. GO annotation was retrieved from the org.Hs.eg.db Bioconductor package version 3.13.0 (unique entrez gene identifers from the org. Hs.egGO2ALLEGS object)23. Mapping between GO identifers and GO terms was taken from GO.db version 3.13.024. Among all GO terms, biological processes with at least 5 and at most 1000 genes were considered. Gene symbols were mapped to enter gene identifers using the same version of org.Hs.eg.db23. Gene set enrichment analysis was performed using generally applicable gene-set enrichment (GAGE)25. For this, all genes were ranked based on their multiple testing adjusted p-values and direction of regulation so that most signifcantly up-regulated genes are on the top of the ranked list and most signifcant down-regulated at the bottom. Within GAGE, Wilcoxon-Mann–Whitney tests were chosen as enrichment tests for each direction (GO terms enriched in induced gene expression and suppressed gene expression). Enrichment tests p-values were also adjusted using the Benjamini–Hochberg method21. All analyses were done in R version 4.1.0 (2021–05-18)26. Gene lists (Supplementary table 1, 2) associated with each GO term were scanned for top 10 up-regulated (supplementary table 1) and down-regulated genes (supplementary table 2) identifed by diferential gene expression analysis. And also, for recurrent genes in GO terms top up-regulated or down-regulated genes. Protein–protein interaction analysis and transcription factor prediction Te free online tool Search Tool for the Retrieval of Interacting genes (STRING) (https://string-db.org/) was used for protein–protein interaction (PPI) analysis of the diferentially expressed genes. Te function “Proteins with values” were used with the log2 fold change values received in paired analysis described above, which was then analyzed by use of a normal geneset analysis. Tis was performed on down-regulated genes obtained with model 3. A medium required interaction score of 0.4 was used and no limitations on the number of interactions was set. Disconnected proteins was hidden in the exported network27. PPI analysis was used to identify hub genes. Te ChEA3 data base was used to identify transcription factors related to the diferentially expressed genes. ChEA3 is a transcription factor enrichment tool designed to rank transcription factors in user-submitted gene sets. Te mean rank method was used28. Target genes of selected transcription factors where then further investigated by use of information from https://genecards.org

Comments on the Quality of English Language

-

Author Response

                                                                                                                             Bergamo, August 1,2024

Dear      

Editor-in-Chief

IJMS

Please find attached the revised manuscript entitled “In Situ Analyses of Placental Inflammatory Response to SARS-CoV-2 Infection in Cases of Mother–Fetus Vertical Transmission” by Denise Morotti, Silvia Tabano, Gabriella Gaudioso, Tatjana Radaelli, Giorgio Alberto Croci, Nicola Bianchi, Giulia Ghirardi, Andrea Gianatti, Luisa Patanè, Valeria Poletti de Chaurand, David A. Schwartz, Mohamed A.A.A. Hagazi and Fabio Grizzi, which we would like to submit for publication in IJMS as Original Article.

First at all we would like to thank the Reviewers for their feedback and their precious suggestion and the Editorial Committee. We have included in the manuscript all the requested amendments highlighted in red.

Detailed replies to the Reviewers’ comments are provided below.

Comments from Reviewer 1#

R. Abstract: Transcriptomics analysis revealed significant differences in gene expression between the placentae of COVID-19-affected mother/newborn pairs and controls. Affected placentae showed significant changes in 334 genes (p <0.05) or 334 genes (p<0.01). A reproducible statistical analysis plan should be included in the Methods section to support these findings and other findings reported in the abstract.

A.Thank you for the valuable comment. We have added a dedicated paragraph titled “Statistical Analysis” to thoroughly explain all the statistical methods employed in this study.

R. There appear to be multiple potential sources of correlation in the data (placentas from mother-newborn pairs; fetal and maternal sides of the placenta; fetal side, intermediate zone and maternal side of the placenta; regions of interest; and twins). This potential lack of independence in the data is important to address through inference or hypothesis testing. The Benjamini-Yekutieli false discovery rate method accounts for this expectation that significant differences or changes in genes may be correlated with or dependent on each other.

A.Thank you for the valuable comment. In the revised manuscript, we have included a detailed and appropriate statistical analysis of all findings. Additionally, two new sections have been added: "Statistical Analysis" and "Bioinformatics Analysis of Differentially Expressed Genes."

R. RNAscope 31 technology, combined with image analysis, revealed differing mRNA levels of ACE2, SARS-CoV-2, 32 IL1-β, IL-6, IL-8, IL-10, and TNF-α between the maternal and fetal sides of the placenta. Please address the statistical methods used for these comparisons.

A.Thank you for the valuable comment. We used the RNAscope hybridization technique to detect SARS-CoV-2, ACE2, IL1-β, IL-6, IL-8, IL-10, and TNF-α mRNA in formalin-fixed, paraffin-embedded placenta tissue. The reactive surface area was quantified in three distinct regions of interest (ROIs) on the maternal, intermediate, and fetal sides of the placenta using a computer-aided image analysis system that identifies brown chromogenic reactivity through RGB color segmentation. Statistical differences in RNA coverage among the different placenta zones were assessed using Student's t-test. Additionally, correlation analysis has been included in the revised manuscript.

R. Bioinformatic analysis of differentially expressed genes should also be addressed.

A. Thank you for the valuable comment. In the revised manuscript, we have included a detailed new section titled "Bioinformatics Analysis of Differentially Expressed Genes" in the Methods. The Results section has been expanded with separate paragraphs explaining the bioinformatic analysis performed.

R. Limitations of gene expression analysis might include technical variability that can affect the reproducibility of results.

A. Thank you for the valuable comment. In the revised manuscript, we have discussed the limitations of gene expression analysis in clinical sciences.

It is true that RNA and protein expression profiling and other technologies have revolutionized how clinicians study the molecular basis of human pathologies and drug effects. These technologies promise efficient, high-throughput methods to delineate mechanisms of action and predict disease progression. However, achieving this requires understanding methodological constraints to design genome-scale studies and interpret large data sets. Recent cancer genetics experiments have shown gene expression profiling can accurately classify tumor phenotypes, offering hope for its predictive potential. Despite these expectations, it remains uncertain how gene expression profiling will enhance our understanding of disease mechanisms and realize its full potential. The complexity of human biology, non-linear dynamics of biological processes, and technical variability such as sample preparation, RNA quality, and platform differences continue to pose significant challenges.

R. Furthermore, comparing gene expression between the fetal and maternal sides of the placenta revealed that all differentially expressed genes were overexpressed on the fetal side. How were these changes in gene expression summarized and analyzed across samples (fold-change estimates, ratios). Discuss replicates and appropriate statistical tests and multiple testing correction.

A. Thank you for the observation. Since this experiment is outside the scope of our research and is based on preliminary findings, we believe the sentence in the Results section should be removed. Consequently, this analysis has not been included in the manuscript.

R. In the following sections, we discuss creating ratios and fold-change estimates, and for those experiments with sufficient biological replicates, we discuss appropriate statistical tests and multiple testing correction.

A. Thank you for the valuable comment. In the revised manuscript, we have included a detailed and appropriate statistical analysis of all findings. Additionally, two new sections have been added: "Statistical Analysis" and "Bioinformatics Analysis of Differentially Expressed Genes."

R. Figure 2 shows SARS-CoV-2 RNA in fetal side, intermediate zone and maternal side of the placenta.

A. Thank you. Figure 2 shows the results of the RNAscope technique used to detect SARS-CoV-2 mRNA in formalin-fixed, paraffin-embedded placental section. Photographs A, B, and C correspond to the fetal side, intermediate zone, and maternal side of the placenta, respectively. The legend has been improved for clarity.

R. Figure 3A graphically represents the distribu- 119 tion of RNA-covered surfaces in the fetal side, intermediate zone, and maternal side of the 120 placenta for each investigated case. Figure 3B illustrates the gradients based on the mini- 121 mum and maximum amounts of single RNA probes observed across all investigated 122 cases. With the exception of a twin (Case 3A),

A. Thank you. The text and the legend have been improved for clarity.

R. For each slide, three ROIs were taken at the same size (90mm2 each). 356 These ROIs represented maternal and fetal side of the placenta plus another intermediate 357 region. To calculate the gradient (G) of expression between the maternal and fetal sides of 358 the placenta, we utilized the following formula,

A.Thank you. The text has been clarified in the Methods, Results, and Discussion sections.

R. transcriptomics analysis aimed at examining differences in gene expression between placentas from mother-newborn pairs affected by COVID-19 and those from unaffected controls.

A. Thank you. In the revised manuscript, we have included a detailed and appropriate statistical analysis of all findings. Additionally, two new sections have been added: "Statistical Analysis" and "Bioinformatics Analysis of Differentially Expressed Genes."

R. Additionally, it investigates the in-situ expression of molecules involved in placental inflammation, providing insights into how the coronavirus potentially traverses the maternal-fetal interface.

A. Thank you. The text has been clarified in the Results and Discussion sections.

 R. A heatmap showing the spatial distribution of COVID mRNA, displaying a gradient from 154 the fetal zone (F) of the placenta towards the maternal side (M) in Case 1. The values within the 155 subregions represent the percentage of mRNA-covered surface

A. Thank you. The text has been clarified in the Results and Discussion sections. The coefficient of variability has been also added.

R. Additionally, genes involved in cell signaling (e.g., CCL20, C3, MARCO) and immune response (e.g., 172LILRA3, CXCL10, CD48, CD86) were upregulated in group 1. High expression of IL1RN, IL18, and LILAR3 in group 1 suggests their role in the inflammatory response to SARS-CoV-2.

A. Thank you. The text has been clarified in throughout the manuscript.

R. With RNAscope technology, combined with image analysis, we observed varying percentages of RNA-covered surfaces between the maternal and fetal sides of the placenta (Figure 2 and 3). Specifically, SARS-CoV-2 RNA showed a gradient with higher expression on the fetal side of the placenta (Figure 3A).

A. Thank you. The text has been clarified in the Methods, Results, and Discussion sections.

R. Bioinformatic analysis of differentially expressed genes

Gene filtering and data transformation was performed. Genes with very low expression were excluded (less than 3 samples with at least 10 counts). Gene expression data quality was assessed using variance mean dependency plots, heatmap of euclidean distances between samples and centered principal component analyses. All differential expression analysis was performed on the raw count data from these 20 samples afer filtering out low expression genes. DESeq2 version 1.32.0 was used for differential expression analyses.

Given that a filtering based on low expression was already performed, no additional independent fltering was done in DESeq2. P-values were adjusted by using the Benjamini–Hochberg method21 and genes were considered statistically significant, if adjusted p-values was lower than 0.05. Te frst diferential expression was paired by mother 3 Vol.:(0123456789) Scientifc Reports | (2024) 14:4210 | https://doi.org/10.1038/s41598-024-54724-3 www.nature.com/scientificreports/ identifer with no other adjustment variables in model 1. Te analysis was further adjusted by mode of delivery (categorized as spontaneous delivery vs. instrumental delivery which included deliveries with vacuum extraction, cesarean section or emergency cesarean-section.) in model 222. Likelihood ratio tests were performed to assess whether interaction terms should be added in the models. Tese likelihood ratio tests compared models with and without interaction terms and 0.053% of the genes (8 genes) had adjusted p-values below 0.05 (p-value distribution close to uniform) so it was decided to not have interaction terms in the models. Te last model 3 was adjusted for delivery mode, maternal BMI and gestational age at birth provided as continuous variables in days22. No likelihood ratio tests were performed on these fully adjusted models. For top signifcant genes of each analysis, normalized counts (variance stabilizing transformed data) were plotted for visualizing results. A Gene Ontology (GO) enrichment analysis was performed on the unadjusted model 1 and fully adjusted model 3 analysis adjusting for delivery mode, maternal BMI, and gestational age. GO annotation was retrieved from the org.Hs.eg.db Bioconductor package version 3.13.0 (unique entrez gene identifers from the org. Hs.egGO2ALLEGS object)23. Mapping between GO identifers and GO terms was taken from GO.db version 3.13.024. Among all GO terms, biological processes with at least 5 and at most 1000 genes were considered. Gene symbols were mapped to enter gene identifers using the same version of org.Hs.eg.db23. Gene set enrichment analysis was performed using generally applicable gene-set enrichment (GAGE)25. For this, all genes were ranked based on their multiple testing adjusted p-values and direction of regulation so that most signifcantly up-regulated genes are on the top of the ranked list and most signifcant down-regulated at the bottom. Within GAGE, Wilcoxon-Mann–Whitney tests were chosen as enrichment tests for each direction (GO terms enriched in induced gene expression and suppressed gene expression). Enrichment tests p-values were also adjusted using the Benjamini–Hochberg method21. All analyses were done in R version 4.1.0 (2021–05-18)26. Gene lists (Supplementary table 1, 2) associated with each GO term were scanned for top 10 up-regulated (supplementary table 1) and down-regulated genes (supplementary table 2) identifed by diferential gene expression analysis. And also, for recurrent genes in GO terms top up-regulated or down-regulated genes. Protein–protein interaction analysis and transcription factor prediction Te free online tool Search Tool for the Retrieval of Interacting genes (STRING) (https://string-db.org/) was used for protein–protein interaction (PPI) analysis of the diferentially expressed genes. Te function “Proteins with values” were used with the log2 fold change values received in paired analysis described above, which was then analyzed by use of a normal geneset analysis. Tis was performed on down-regulated genes obtained with model 3. A medium required interaction score of 0.4 was used and no limitations on the number of interactions was set. Disconnected proteins was hidden in the exported network27. PPI analysis was used to identify hub genes. Te ChEA3 data base was used to identify transcription factors related to the diferentially expressed genes. ChEA3 is a transcription factor enrichment tool designed to rank transcription factors in user-submitted gene sets. Te mean rank method was used28. Target genes of selected transcription factors where then further investigated by use of information from https://genecards.org

A. Thank you for the suggestion. We hypothesize that the above text serves as a reference guide, likely copied from a previously published manuscript. In the revised manuscript, we have included a detailed and appropriate statistical analysis of all findings. Additionally, two new sections have been added: "Statistical Analysis" and "Bioinformatics Analysis of Differentially Expressed Genes." In addition, we have introduced the above research in the References section.

In consideration of IJMS taking action in reviewing and editing this submission, the authors hereby transfer, assign or otherwise convey all copyright to IJMS in the event that the paper is published in IJMS.

We hope you will find the manuscript of sufficient merit and interest to warrant its publication in your Journal.

Please for any other information do not hesitate to contact me.

Sincerely,

Denise Morotti

Reviewer 2 Report

Comments and Suggestions for Authors

In 3 cases of vertical transmissions of SARS-CoV-2, the authors studied the transcriptomics differences in gene expression between their placentas and the ones of unaffected controls, also analysed the expression of molecules involved in placental inflammation.

The article is interesting and help to understand how the placenta is affected and its possible mechanism.

Minor questions

Were the placental of the unaffected controls paired for gestational age?

How the authors explain the heterogeneity of some findings? Could the severity of the maternal disease, the gestational age or any other factor play any role.

How the authors explain the negative first nasopharyngeal (NP) swab in case number 2.  

Author Response

                                                                                                                             Bergamo, August 1,2024

Dear      

Editor-in-Chief

IJMS

Please find attached the revised manuscript entitled “In Situ Analyses of Placental Inflammatory Response to SARS-CoV-2 Infection in Cases of Mother–Fetus Vertical Transmission” by Denise Morotti, Silvia Tabano, Gabriella Gaudioso, Tatjana Radaelli, Giorgio Alberto Croci, Nicola Bianchi, Giulia Ghirardi, Andrea Gianatti, Luisa Patanè, Valeria Poletti de Chaurand, David A. Schwartz, Mohamed A.A.A. Hagazi and Fabio Grizzi, which we would like to submit for publication in IJMS as Original Article.

First at all we would like to thank the Reviewers for their feedback and their precious suggestion and the Editorial Committee. We have included in the manuscript all the requested amendments highlighted in red.

Detailed replies to the Reviewers’ comments are provided below.

Comments from Reviewer 2#

In 3 cases of vertical transmissions of SARS-CoV-2, the authors studied the transcriptomics differences in gene expression between their placentas and the ones of unaffected controls, also analysed the expression of molecules involved in placental inflammation.

The article is interesting and help to understand how the placenta is affected and its possible mechanism.

Minor questions

R. Were the placental of the unaffected controls paired for gestational age?

A. Thank you for your question. Infected and control placentae were matched for gestational age.

R. How the authors explain the heterogeneity of some findings? Could the severity of the maternal disease, the gestational age or any other factor play any role.

A. Thank you for your important question. This research is based on the study of four cases. We acknowledge that the small sample size highlights intrinsic biological variability. Additionally, we used the term "heterogeneity" to describe a non-uniform behavior or magnitude of a variable. For instance, coupling RNAscope technology with a computer-aided image analysis system provides a quantitative measurement specific to a two-dimensional tissue section, capturing a snapshot of a more complex, dynamic process. Furthermore, viral load is a parameter that significantly influences the severity of the disease and its pathological phenotype.

R. How the authors explain the negative first nasopharyngeal (NP) swab in case number 2.

A. Thank you for your observation. Given that all subjects tested positive for SARS-CoV-2, we attribute the initial negative test to the minor viral load of case 2, may be the baby turned positive in few hours. Therefore, we consider this initial result to be a “false negative”.

In consideration of IJMS taking action in reviewing and editing this submission, the authors hereby transfer, assign or otherwise convey all copyright to IJMS in the event that the paper is published in IJMS.

We hope you will find the manuscript of sufficient merit and interest to warrant its publication in your Journal.

Please for any other information do not hesitate to contact me.

Sincerely,

Denise Morotti

Round 2

Reviewer 1 Report

Comments and Suggestions for Authors

See attached review.

Author Response

Bergamo, August 8, 2024

Dear      

Reviewer 1

Please find attached the revised manuscript entitled “In Situ Analyses of Placental Inflammatory Response to SARS-CoV-2 Infection in Cases of Mother–Fetus Vertical Transmission” by Denise Morotti, Silvia Tabano, Gabriella Gaudioso, Tatjana Radaelli, Giorgio Alberto Croci, Nicola Bianchi, Giulia Ghirardi, Andrea Gianatti, Luisa Patanè, Valeria Poletti de Chaurand, David A. Schwartz, Mohamed A.A.A. Hagazi and Fabio Grizzi, which we would like to submit for publication in IJMS as Original Article.

First at all we would like to thank the Reviewer for his/her feedback and their precious suggestion. We have included in the manuscript all the requested amendments highlighted in red.

Detailed replies to the Reviewers’ comments are provided below.

Reviewer 1

Abstract: The current study focuses on a transcriptomics analysis aimed at examining differences in gene expression between placentas from mother-newborn pairs affected by COVID-19 and those from unaffected controls. Please report the sample size for the controls.

Thank you for your question. The number of samples used as control (n = 2) has been reported in the Abstract and Materials and Methods sections. Infected and control placentae were matched for gestational age.

Abstract: A non-statistically significant gradient for SARS-CoV-2 was observed, with a higher surface coverage on the fetal 34 side (2.42 ± 3.71%) compared to the maternal side (0.74 ± 1.19%) of the placenta. Please clarify the statistics reported (2.42 ± 3.71%, mean ± SD). Is the difference observed potentially biologically important but the statistical power low?

Thank you for the observation. We analyzed four cases of infection. In three of these cases, we found a higher concentration of SARS-CoV-2 RNA on the fetal side of the placenta compared to the maternal side, with the exception of one tween. However, this difference was not statistically significant. Despite the potential biological importance of these findings, the small sample size limits our ability to draw definitive conclusions. Therefore, further research is necessary to confirm our observations.

Statistical analysis: The data were expressed as mean ± standard deviation. Statistical differences among variables were analyzed by t-test. Please be specific when describing the statistical methods used to compare differences within a group (paired differences using a two-sided, paired-difference t-test). I assume differences between groups (controls versus COVID-19) were compared for each specific outcome with a two-sided, two-sample equal-variance t-test. Please clarify.

Thank you for your question. We used an unpaired two-tailed t-test. This has been clarified into the statistical analysis paragraph.

Table 1. Is it possible to report 95% confidence intervals for log2 fold change?

Thank you for your observation. We have included the 95% confidence intervals for log2 fold change in Table 1.

Table 2: Report the mean and standard deviation. Define Q value in the analysis plan.

Thank you for your observation. We have included the mean and standard deviation in Table 2. Additionally, the definition of the Q value (i.e., the minimum false discovery rate at which an observed score is considered significant) is provided in a note below Table 2.

I had intended for my original review to be limited to the following paragraphs. The remainder of the review was inadvertently included in the review. Apologies.

Thank you for the requested revisions to the initial manuscript, which have been taken into account to enhance its quality.

Abstract: Transcriptomics analysis revealed significant differences in gene expression between the placentae of COVID-19-affected mother/newborn pairs and controls. Affected placentae showed significant changes in 334 genes (p <0.05) or 334 genes (p<0.01). A reproducible statistical analysis plan should be included in the Methods section to support these findings and other findings reported in the abstract.

Thank you. We have included in the revised version of the manuscript two paragraphs aimed at explaining the statistical analysis and the bioinformatic analysis.

There appear to be multiple potential sources of correlation in the data (placentas from mother- newborn pairs; fetal and maternal sides of the placenta; fetal side, intermediate zone and maternal side of the placenta; regions of interest; and twins). This potential lack of independence in the data is important to address through inference or hypothesis testing. The Benjamini-Yekutieli false discovery rate method accounts for this expectation that significant differences or changes in genes may be correlated with or dependent on each other.

Thank you. In the revised version of the manuscript, we have included three new paragraphs that explain the statistical analysis, the bioinformatic analysis, and the coupled RNAscope technology with computer-aided image analysis for evaluating RNA-covered surfaces on FFPE tissues (RNAscopeÒ image analysis).

RNAscope 31 technology, combined with image analysis, revealed differing mRNA levels of ACE2, SARS-CoV-2, 32 IL1-β, IL-6, IL-8, IL-10, and TNF-α between the maternal and fetal sides of the placenta. Please address the statistical methods used for these comparisons. Bioinformatic analysis of differentially expressed genes should also be addressed. Limitations of gene expression analysis might include technical variability that can affect the reproducibility of results.

Thank you. In the revised version of the manuscript, we have included three new paragraphs that explain the statistical analysis, the bioinformatic analysis, and the coupled RNAscope technology with computer-aided image analysis for evaluating RNA-covered surfaces on FFPE tissues (RNAscopeÒ image analysis).

Furthermore, comparing gene expression between the fetal and maternal sides of the placenta revealed that all differentially expressed genes were overexpressed on the fetal side. How were these changes in gene expression summarized and analyzed across samples (fold-change estimates, ratios). Discuss replicates and appropriate statistical tests and multiple testing correction.

Thank you. As previously reported the revised version, since this experiment is outside the scope of our research and is based on preliminary findings, we believe the sentence in the Results section should be removed. Consequently, this analysis has not been included in the manuscript.

We hope you will find the manuscript of sufficient merit and interest to warrant its publication in IJMS.

Please for any other information do not hesitate to contact me.

Sincerely,

Denise Morotti